# Trends in Diabetes-Related Potentially Preventable Hospitalizations in Adult Population in Spain, 1997–2015: A Nation-Wide Population-Based Study

**DOI:** 10.3390/jcm8040492

**Published:** 2019-04-11

**Authors:** Ricardo Gómez-Huelgas, Carmen M. Lara-Rojas, María D. López-Carmona, Sergio Jansen-Chaparro, Raquel Barba, Antonio Zapatero, Ricardo Guijarro-Merino, Francisco J. Tinahones, Luis M. Pérez-Belmonte, M. Rosa. Bernal-López

**Affiliations:** 1Servicio de Medicina Interna, Hospital Regional Universitario de Málaga, Instituto de Investigación Biomédica de Málaga (IBIMA), Universidad de Málaga (UMA), 29010 Málaga, Spain; ricardogomezhuelgas@hotmail.com (R.G.-H.); carmen_lara_rojas@hotmail.com (C.M.L.-R.); sjansenc@gmail.com (S.J.-C.); rguijarrom@gmail.com (R.G.-M.); robelopajiju@yahoo.es (M.R.B.-L.); 2Centro de Investigación Biomédica en Red Fisiopatología de la Obesidad y Nutrición (CIBERobn), Instituto de Salud Carlos III, 28029 Madrid, Spain; fjtinahones@hotmail.com; 3Servicio de Medicina Interna, Hospital Universitario Rey Juan Carlos, 28933 Móstoles, Madrid, Spain; raquel.barba@hospitalreyjuancarlos.es; 4Servicio de Medicina Interna, Hospital Universitario de Fuenlabrada, Universidad Rey Juan Carlos, 28942 Fuenlabrada, Madrid, Spain; antonio.zapatero@salud.madrid.org; 5Unidad de Gestión Clínica de Endocrinología y Nutrición, Hospital Universitario Virgen de la Victoria, Instituto de Investigación Biomédica de Málaga (IBIMA), Universidad de Málaga (UMA), 29010 Málaga, Spain; 6Centro de Investigación Biomédica en Red Enfermedades Cardiovasculares (CIBERCV), Instituto de Salud Carlos III, 28029 Madrid, Spain

**Keywords:** diabetes complications, diabetes care, diabetes mellitus, potentially preventable hospitalizations

## Abstract

We aimed to assess national trends in the rates of diabetes-related potentially preventable hospitalizations (overall and by preventable condition) in the total adult population of Spain. We performed a population-based study of all adult patients with diabetes who were hospitalized from 1997 to 2015. Overall potentially preventable hospitalizations and hospitalizations by diabetes-related preventable conditions (short-term complications, long-term complications, uncontrolled diabetes, and lower-extremity amputations) were examined. Annual rates adjusted for age and sex were analyzed and trends were calculated. Over 19-years-period, 424,874 diabetes-related potentially preventable hospitalizations were recorded. Overall diabetes-related potentially preventable hospitalizations decreased significantly, with an average annual percentage change of 5.1 (95%CI: −5.6–(−4.7%); *p*_trend_ < 0.001). Among preventable conditions, the greatest decrease was observed in uncontrolled diabetes (−5.6%; 95%CI: −6.7–(−4.7%); *p*_trend_ < 0.001), followed by short-term complications (−5.4%; 95%CI: −6.1–(−4.9%); *p*_trend_ < 0.001), long-term complications (−4.6%; 95%CI: −5.1–(−3.9%); *p*_trend_ < 0.001), and lower-extremity amputations (−1.9%; 95%CI: −3.0–(−1.3%); *p*_trend_ < 0.001). These reductions were observed in all age strata for overall DM-related PPH and by preventable condition but lower-extremity amputations for those <65 years old. There was a greater reduction in overall DM-related PPH, uncontrolled DM, long-term-complications, and lower extremity amputations in females than in males (all *p* < 0.01). No significant difference was shown for short-term complications (*p* = 0.101). Our study shows a significant reduction in national trends for diabetes-related potentially preventable hospitalizations in Spain. These findings could suggest a sustained improvement in diabetes care in Spain, despite the burden of these diabetes-related complications and the increase in the diabetes mellitus prevalence.

## 1. Introduction

Patients with diabetes mellitus (DM) are at greater risk to be hospitalized. Admission rates for these patients have been reported as being up to 6 times higher than those of patients without DM [1,2]. A high proportion of these hospitalizations are due to DM complications both in patients with type-1 and type-2 DM [1,2]. These hospitalizations and its associated complications are costly, ranging from one-third to one-half of the total direct medical expenditure for DM in developed countries [3,4].

A substantial proportion of hospitalizations for acute or chronic conditions in relationship with DM may be potentially avoided with an adequate and effective ambulatory care. Early diagnosis, effective treatment, and appropriate education could improve the management of DM, prevent the development and worsening of DM-related complications, and reduce hospitalizations [5]. These DM-related potentially preventable hospitalizations (PPH) have been identified as a good indicator of the efficiency and quality of health system [6,7,8]. The Agency for Health Care Research and Quality (AHRQ) has identified four conditions of DM-related PPH: short-term DM complications, long-term DM complications, uncontrolled DM, and lower-extremity DM-related amputations [9].

This issue has been controversially described in United States and European countries, with declining and increasing rates of DM-related PPH in the last two decades [3,5,10,11,12,13,14]. The aim of this study was to assess national trends in the rates of DM-related PPH (overall and by preventable hospitalization condition) in the entire adult population in Spain from 1997 to 2015, broken down by age and sex.

## 2. Materials and Methods

### 2.1. Study Design

Using the Spanish National Hospital Discharge Database (CMBD, Conjunto Mínimo Básico de Datos) from the Spanish National Health System, we performed a nation-wide, retrospective, population-based study of all adult patients with DM who were hospitalized in Spain between 1997 and 2015. Overall DM-related PPH and DM-related PPH conditions were analyzed.

The CMBD compiles patient data (sex, birth date, admission and discharge dates, readmissions, up to 20 procedures performed during the hospital stay, and up to 14 discharge diagnoses) for all public and private hospitals in Spain, covering more than 98% of hospitalizations. Admissions in emergency departments lasting less than 24 h are not included in the database. This national database is coordinated by the Spanish Ministry of Health, Social Services and Equality, which sets standards for record keeping and performs routine audits of the data [15].

Disease classifications were established according to the International Classification of Diseases, Ninth Revision, Clinical Modification (ICD-9-CM), which is used in the Spanish CMBD. We selected all patients aged 18 or older who were discharged from January 1997 to December 2015 with the diagnosis of DM (ICD-9-CM codes: 250.0–250.9).

DM-related PPH conditions were defined according to AHRQ report for Prevention Quality Indicators 1, 3, 14 and 16 (ICD-9-CM codes) [16]: DM short-term complications (ketoacidosis, hyperosmolarity, or coma. ICD-9-CM codes: 250.1–250.3), DM long-term complication (renal, eye, neurological, circulatory, or complications not otherwise specified. ICD-9-CM codes: 250.4–250.9), uncontrolled DM without mention of short- or long-term complications (high glucose concentrations. ICD-9-CM codes: 250.02 or 250.03), and procedures for lower-extremity amputation (ICD-9-DM codes: 84.10 and 84.12–84.19).

Data confidentiality and patient anonymity were maintained at all times, in accordance with Spanish regulations on observational studies [16]. Patient identifying information was deleted before the database was analyzed. It is not possible to identify patients on an individual level either in this article or in the database. Due to the anonymous nature and mandatory collection of the information included in the dataset, informed consent from the patients was not necessary.

### 2.2. Statistical Analysis

All data analysis were performed using SPSS Statistics for Windows, version 15.0 (SPSS Inc., Chicago, IL, USA), Joinpoint Regression Program, version 4.2.0.2 (Statistical Methodology and Applications Branch, Surveillance Research Program, National Cancer Institute, Bethesda, MD, USA) and Epidat: programa para análisis epidemiológico de datos, version 4.2 (Consellería de Sanidade, Xunta de Galicia, A Coruña, Galicia, España).

The comparative analysis was assessed by carrying out repeated measures analysis of variance (ANOVA).

Annual rates for overall DM-related PPH and each DM-related preventable condition were calculated per 1000 hospitalized patients with DM, adjusting for age and sex using the direct method. The total Spanish population as of 2015 was used as the standard population. Also, data was stratified by age (≤18, 19–44, 45–64, 65–74, and ≥75 years old) and sex (male and female). The average annual percent change (AAPC) and 95% confidence interval (CI) was computed for overall DM-related PPH and by preventable DM-related condition and adjusted by age and sex. The trends were considered significant when the slope of the trend was not equal to zero and a *p*-value was <0.05.

## 3. Results

Among the 7,393,341 admissions in patient with DM in Spain during the 19-years study period in Spain, 5.7% (*n* = 424,874) were DM-related PPH.

The adjusted number of overall DM-related patients per 1000 hospitalized patients with DM decreased from 1997 to 2015 (Figure 1).

Over the study period, the adjusted rates of DM-related PPH according to preventable condition decreased (Figure 2). Short-term complications were the most frequent cause of DM-related PPH, followed by long-term complications, uncontrolled DM, and lower-extremity amputations.

The AAPC for overall DM-related PHH and according to preventable condition for both the overall diabetic population hospitalized and stratified by age and sex are summarized in Table 1. Overall DM-related PPH experienced a significant average annual reduction over 19-year-period. In regards to preventable condition, the greatest average decrease was observed in uncontrolled DM, followed by short-term complications, long-term complications, and lower-extremity amputations. In addition, these reductions were observed in all age strata for overall DM-related PPH and by preventable condition but lower-extremity amputations for those <65 years old. There was a greater reduction in overall DM-related PPH, uncontrolled DM, long-term-complications and lower extremity amputations in females than in males (all *p* < 0.01). Hospitalizations for short-term complications showed non-significant difference (*p* = 0.101).

The AAPC for overall DM-related PHH and according to preventable condition for both the overall diabetic population hospitalized and stratified by age and sex are summarized in Table 1. Overall DM-related PPH experienced a significant average annual reduction over 19-year period. In regards to preventable condition, the greatest average decrease was observed in uncontrolled DM, followed by short-term complications, long-term complications, and lower-extremity amputations. In addition, these reductions were observed in all age strata for overall DM-related PPH, short-term and long-term complications, and uncontrolled diabetes. Hospitalization reduction for lower-extremity amputations was only observed for those ≥65 years old. There was a greater reduction in overall DM-related PPH, uncontrolled DM, long-term-complications and lower extremity amputations in females than in males (all *p* < 0.01). No significant difference was shown for short-term complications (*p* = 0.101).

## 4. Discussion

This Spanish population-based retrospective study of all patients with DM who were hospitalized, found that adjusted overall DM-related PPH and according to preventable condition decreased from 1997 to 2015. Short-term and long-term complications remained as the most frequent conditions of PPH during the period of study, followed by uncontrolled DM and lower-extremity amputations. The greatest average annual decrease was observed in uncontrolled DM, followed by short-term complications, long-term complications, and lower-extremity amputations. The decrease in hospitalizations for lower extremity amputations was non-significant among patients <65 years old. Females experienced a greater reduction overall in uncontrolled diabetes, long-term complications, and lower extremity amputation hospitalizations.

These results are important because they show decreasing DM-related PPH rates in a large sample over a wide period of time. Furthermore, they provide valuable information such as the adjusted AAPC and stratification by age and sex. Also, although there is a considerable interest in health care aspects of DM, epidemiological data of hospitalizations in patients with DM have been insufficiently described in many regions worldwide.

Prior studies from different countries have reported similar DM-related PPH trends. In the United States of America, a nationwide study published by Wang et al. [3], using the Health-care Cost and Utilizations Project National Inpatient Sample (NIS) in order to obtain data about hospitalizations, reported a significant decrease for all DM-related preventable conditions, except for short-term complications, during the period 1998–2006. Uncontrolled DM, similarly to our findings, showed the greatest reduction. However, short-term complications showed a non-significant decrease, mainly attributable to the insignificant decrease among the youngest patients (<45 years old). It was not clear why there was non-significant decline in this group of patients. In our study, every DM-related preventable condition experienced a higher reduction in older age-groups, especially among those patients who are 75 years old or above. Very few studies have explored age and sex differences in DM-related PPH [3,11] but all of them have showed similar trends to our study.

Decline in hospitalization rates for uncontrolled DM was also reported by AHRQ from 1994 to 2000 [17]. However, in contrast to our results, the AHRQ study showed an increase for short-term complications, and no changes for both long-term complications and lower-extremity amputations.

Recently, a study found that DM-related PPH rates did not change significantly during the years 2005–2014 in the United States, unlike the results in a previous reports. A slight increase in hospitalization rates due to short-term complications balanced by a slight decrease in hospitalization rates due to uncontrolled diabetes has been suggested as the main reason of these trends [10].

In Europe, several studies have reported data in this line. A retrospective review of medical records of patients hospitalized for acute diabetic complications carried out in Italy between 2001 and 2010 showed a decreasing temporal trend [11]. Consistently with these data, a recently published study conducted in Finland from 1996 to 2011 showed a general decline in the incidence of DM-related PPH [12]. Another study focused on diabetic ketoacidosis showed increasing rates in hospital admissions between 1999 and 2010 in Wales [13]. In Spain, studies reported by López-de-Andrés in recent years have examined trends in non-traumatic lower-extremity amputations in patients with type-1, type-2 and without DM in two periods, 2001–2008 and 2001–2012 in Spain [14,18]. Using the same national database than in our study, they reported a decrease for lower-extremity amputation rates in type-1 DM but a significant increase in patients with type-2 DM.

The different methodologies used to estimate the hospitalization rates in each study and different periods analyzed could be the main reasons for these differences. In our study, we used the total number of adults who were hospitalized with DM, adjusting concomitantly by age and sex with the direct method, and the total Spanish population of 2015 as the standard population, being probably the most sensible method to determine changes in the rates of DM-related complications.

Although both males and females showed a significant reduction in the overall and specific conditions of DM-related PPH, there was a higher reduction among females for all conditions except for short-term complications (*p* = 0.101) over the study period. These trends have also been observed in other studies [3,11,13], but the reasons for these differences have not been widely debated. In our point of view, these results could suggest that short-term complications would be closely related to the management of the disease and long-term complications, uncontrolled DM, and lower-extremity amputations, in addition to glycemic control, would be associated with gender-specific risk factors, including sex hormone differences and sociocultural factors such as different forms of nutrition, life styles, or stress.

Reasons for the reductions in rates of DM-related PPH are not fully understood. Improvements in primary care, the availability of safer and more effective antidiabetic drugs and efforts of clinicians and health organizations in prevention or earlier detection strategies in DM could be the main reasons for these declines [3,12,19,20]. Our findings may reflect an overall improvement in DM care in Spain in recent decades, and are consistent with the recent evidence about the improvement of glycemia and cardiovascular risk in patients with DM observed in Spain [21]. In addition, these reductions in trends for DM-related PPH in Spain may have important clinical implications in terms of reducing complications, mortality, and the cost associated with the hospitalizations.

The persistent high rates of smoking among patients with DM in Spain and the funding of new services for diabetic foot disease have increased the patients who are diagnosed with peripheral artery disease and subsequently, the identification of need for vascular procedures [19,22]. These could be the main reasons for the lack of a significant decline in lower-extremity amputations in patients under 65 years-old.

DM-related PPH are an emerging health care outcome to health systems worldwide [23,24]. The results of several international studies have shown reductions in DM-related preventable conditions when multidisciplinary and integrated care units for patients with DM have been implemented as well as improvements in health care expenditure [25,26]. Further epidemiological studies and interventional preventive programs aimed to reduce overall and specific rates in DM-related PPH should be conducted in many developed countries around the world.

We acknowledge the following limitations in this study. Firstly, using ICD-9-CM codes to describe trends may lead to bias and inaccuracy, depending on the validity of the coding. In this regard, we could not distinguish between type-1 and type-2 DM, as diagnosis codes are not reliable. Secondly, we did not study geographical variations in the rate of DM-related PPH within the same country and it could be important. Studies from Italy [11], Finland [12], Canada [27], and the United States of America [28] have reported marked geographical variations in the rate of DM-related PPH. The accessibility to DM care in certain areas, especially in primary care, has been suggested as the main reason of these differences. Similarly, it has been reported that social and individual disparities play a role in the frequency of DM-related PPH, being more common among individuals with intellectual and developmental disability [29], some racial groups [30,31], and those with low or medium incomes [32]. Again, the role of these factors has not been evaluated in our study since this type of information is not collected in the database. Finally, it is difficult to compare rates across countries and studies because definitions of disease and methodology may differ.

## 5. Conclusions

This population-based study of all patients with DM who were hospitalized, found that adjusted overall DM-related PPH according to preventable condition decreased from 1997 to 2015. Short-term and long-term complications remained as the most frequent conditions of PPH during the period of study and the greatest average annual decrease was observed in uncontrolled DM. These results could reflect as a good indicator of the improvement in DM care in Spain, despite the burden of these complications due to the increase in the prevalence of DM. However, further research is needed to implement interventions for reducing lower-extremity amputation procedures in the younger strata and to fully identify the reasons for these hospitalization declines, evaluating geographical, gender, social, and individual determinants for the differences in the occurrence of DM-related PPH.

## Figures and Tables

**Figure 1 jcm-08-00492-f001:**
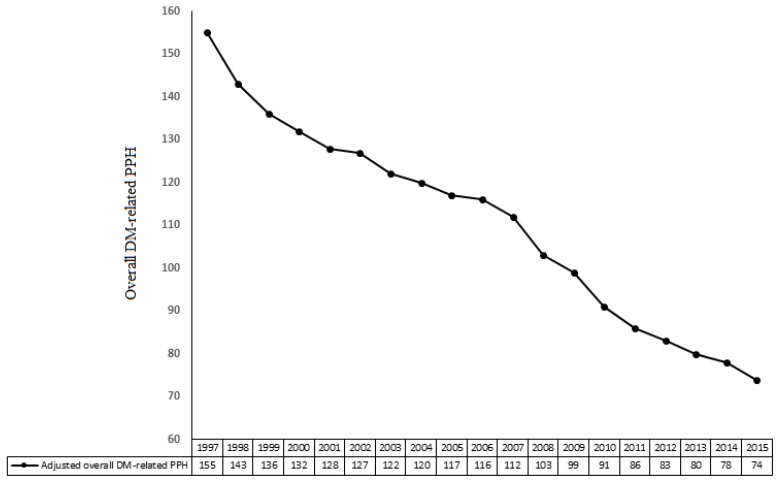
Adjusted number of overall diabetes-related potentially preventable hospitalizations from 1997 to 2015. Data were calculated per 1000 hospitalized patients with diabetes mellitus per year. DM: diabetes mellitus; PPH: potentially preventable hospitalizations.

**Figure 2 jcm-08-00492-f002:**
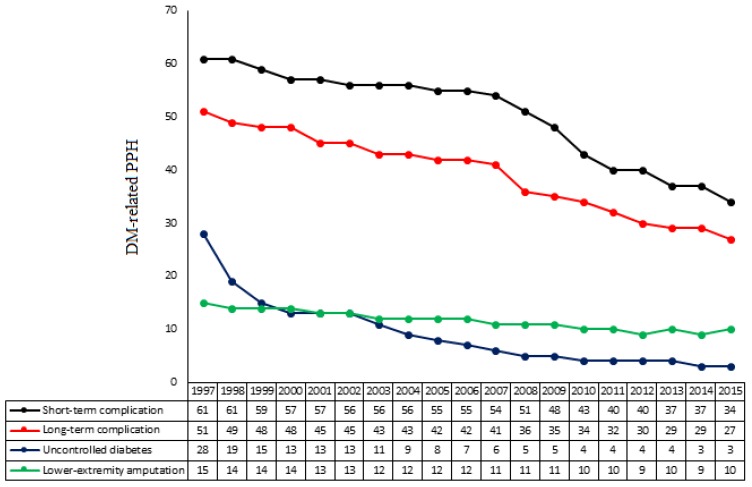
Adjusted number of diabetes-related potentially preventable hospitalizations according to preventable condition from 1997 to 2015. Data were calculated per 1000 hospitalized patients with diabetes mellitus per year. DM: diabetes mellitus; PPH: potentially preventable hospitalizations.

**Table 1 jcm-08-00492-t001:** Annual average percentage change of diabetes-related potentially preventable hospitalizations: overall and stratified by age and sex (1997–2015).

	AAPC	95%CI	*p*-Value (Trend)
Overall DM-related PPH			
18–44 years	−1.9	−2.6 to −1.2	<0.01
45–64 years	−3.4	−4.2 to −3.0	<0.001
65–74 years	−5.2	−6.2 to −5.0	<0.001
≥75 years	−6.0	−6.6 to −5.4	<0.001
Male	−4.9	−5.2 to −4.5	<0.001
Female	−5.8	−6.4 to −5.4	<0.001
Total population	−5.1	−5.6 to −4.7	<0.001
Short-term complications			
18–44 years	−1.9	−2.9 to −1.1	<0.01
45–64 years	−3.7	−4.2 to −3.1	<0.001
65–74 years	−5.7	−6.3 to −5.3	<0.001
≥75 years	−6.0	−6.5 to −5.5	<0.001
Male	−5.5	−5.8 to −5.1	<0.001
Female	−5.1	−5.7 to −4.5	<0.001
Total population	−5.4	−6.1 to −4.9	<0.001
Long-term complication			
18–44 years	−1.9	−2.8 to −1.3	<0.01
45–64 years	−2.6	−3.7 to −2.1	<0.001
65–74 years	−3.9	−4.6 to −3.0	<0.001
≥75 years	−4.2	−4.9 to −3.4	<0.001
Male	−4.0	−4.5 to −3.5	<0.001
Female	−4.8	−5.4 to −4.1	<0.001
Total population	−4.6	−5.1 to −3.9	<0.001
Uncontrolled diabetes			
18–44 years	−2.4	−3.2 to −1.9	<0.001
45–64 years	−3.8	−4.8 to −3.1	<0.001
65–74 years	−5.1	−6.0 to −4.4	<0.001
≥75 years	−6.4	−7.9 to −5.2	<0.001
Male	−6.1	−6.9 to −5.3	<0.001
Female	−7.2	−8.5 to −6.2	<0.001
Total population	−5.6	-6.7 to −4.7	<0.001
Lower-extremity amputations			
18–44 years	−0.2	−0.4 to −0.1	0.111
45–64 years	−0.3	−0.5 to −0.1	0.104
65–74 years	−1.6	−2.7 to −1.2	<0.01
≥75 years	−2.0	−4.0 to −2.1	<0.001
Male	−1.5	−2.9 to −1.2	<0.01
Female	−3.0	−4.0 to −2.4	<0.001
Total population	−1.9	−3.0 to −1.3	<0.01

Total population and stratified by age and sex average annual percentage changes are shown for total and by preventable condition diabetes-related potentially preventable hospitalizations. AAPC: Average annual percentage change; 95%CI: 95% confidence interval.

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
