# Peer review of "Trends in Diabetes-Related Potentially Preventable Hospitalizations in Adult Population in Spain, 1997–2015: A Nation-Wide Population-Based Study"

_jcm, 2019, doi:10.3390/jcm8040492_

Reviewer 1 Report

Re: Trends in diabetes-related potentially preventable hospitalizations in adult population in Spain, 1997-2015: a national-wide population-based study

This study was aimed to assess national trends in the rates of diabetes-related potentially preventable hospitalizations in the diabetic population of Spain. The manuscript was clearly written, but few questions need to be clarified before publication.

1.       Introduction: please provide reference for the sentence “Admission rates for these patients have been reported in up to 6 times higher than those of patients without DM.”

2.       Method: authors used total Spanish population of 2010 as the standard population, but in the discussion, it mentioned that this study used the total Spanish population of 2015 as the standard population.

3.       Result: for the annual average percentage change, the study found the reductions were observed in all age strata for overall DM-related PPH and by preventable condition but lower-extremity amputations for those<65 years old. The later part of the sentence was not clear. Please clearly state that hospitalisation reduction for lower-extremity amputations was not observed for those <65 years="" or="" the="" reduction="" was="" only="" observed="" for="" those="">65 years.

4.       Result: the reduction for uncontrolled DM was greater in males (-7.2) than in females (-6.1) from the data presented in Table 1, which was not consistent with the result in the text. Please double check this part of the result and revise the result in the abstract and discussion section accordingly.

5.       Result: please revise the sentence to “Hospitalisation reduction for short-term complication showed non-significant difference between males and females (p=0.101).”

6.       Discussion: what are the gender-specific risk factors that could explain the observed gender differences?

7.       Discussion: the prevalence of DM and the burden of diabetic complications are still increasing, what are the clinical implications of these findings to the Spain health system? 

Author Response

Thank you for reviewing and considering our manuscript, “Trends in diabetes-related potentially preventable hospitalizations in adult populations in Spain 1997-2015: a national-wide population-based study” for possible publication in the Journal of Clinical Medicine. We greatly appreciate the comments provided to us by the reviewers.

We have made revisions to our manuscript, according the recommendations of the reviewers, and are now resubmitting it to you for your consideration. The reviewers’ comments and our corresponding responses are listed below.

Thank you very much for your kind attention on this matter.

Sincerely

Corresponding authors on behalf of all co-authors.

Luis M. Pérez-Belmonte, MD, PhD

María D. López-Carmona, MD, PhD.

luismiguelpb1984@gmail.com; mdlcorreo@gmail.com

Responses to Reviewers’ Comments:

Reviewer #1:

Trends in diabetes-related potentially preventable hospitalizations in adult population in Spain, 1997-2015: a national-wide population-based study.

This study was aimed to assess national trends in the rates of diabetes-related potentially preventable hospitalizations in the diabetic population of Spain. The manuscript was clearly written, but few questions need to be clarified before publication.

1. Introduction: please provide reference for the sentence “Admission rates for these patients have been reported in up to 6 times higher than those of patients without DM.”

Authors’ reply: We have added this reference.

2. Method: authors used total Spanish population of 2010 as the standard population, but in the discussion, it mentioned that this study used the total Spanish population of 2015 as the standard population.

Authors’ reply: This has been a mistake. We used total Spanish population of 2015 as the standard population. We have changed it in the methods section.

3. Result: for the annual average percentage change, the study found the reductions were observed in all age strata for overall DM-related PPH and by preventable condition but lower-extremity amputations for those<65 years old. The later part of the sentence was not clear. Please clearly state that hospitalisation reduction for lower-extremity amputations was not observed for those <65 years="" or="" the="" reduction="" was="" only="" observed="" for="" those="">65 years.

Authors’ reply: We have clarified this sentence. We have written the following: “In addition, these reductions were observed in all age strata for overall DM-related PPH, short-term and long-term complications, and uncontrolled diabetes. Hospitalization reduction for lower-extremity amputations was only observed for those ≥65 years old.”

4. Result: the reduction for uncontrolled DM was greater in males (-7.2) than in females (-6.1) from the data presented in Table 1, which was not consistent with the result in the text. Please double check this part of the result and revise the result in the abstract and discussion section accordingly.

Authors’ reply: These data was misplaced in the Table 1. Reduction for uncontrolled DM was greater in females (-7.2) than in males (-6.1). We have revised the result in the abstract and discussion section.

5. Result: please revise the sentence to “Hospitalisation reduction for short-term complication showed non-significant difference between males and females (p=0.101).”

Authors’ reply: We have revised this sentence. We have written the following: “No significant difference was shown for short-term complications (p=0,101).”

6. Discussion: what are the gender-specific risk factors that could explain the observed gender differences?

Authors’ reply: We have included the factors that could explain these differences. We have written the following: “…would be associated with gender-specific risk factors, including sex hormones differences and sociocultural factors such as different forms of nutrition, life styles or stress.”

7. Discussion: the prevalence of DM and the burden of diabetic complications are still increasing, what are the clinical implications of these findings to the Spain health system?  

Authors’ reply: We have added a comment about that in the Discussion. The text added was the following: “In addition, these reductions in trends for DM-related PPH in Spain may have important clinical implications in terms of reducing complications, mortality and the cost associated with the hospitalizations."

Reviewer 2 Report

The authors describe a retrospective population-based study of all adult patients with diabetes who were hospitalized from 1997 to 2015 in Spain. Overall potentially preventable hospitalizations and hospitalizations by diabetes-related preventable conditions (short-term complications, long-term complications, uncontrolled diabetes and lower-extremity amputations) were investigated.

Data are well presented and thorough. One of the strengths of this study is the very long time considered in the analysis (19 years) as well as the wide sample of patients enrolled (whole Spanish country).

Among DM long-term complications the authors also cited eye complications. 

Have they also considered data from some diabetic eye screening programs?

Author Response

Thank you for reviewing and considering our manuscript, “Trends in diabetes-related potentially preventable hospitalizations in adult populations in Spain 1997-2015: a national-wide population-based study” for possible publication in the Journal of Clinical Medicine. We greatly appreciate the comments provided to us by the reviewers.

We have made revisions to our manuscript, according the recommendations of the reviewers, and are now resubmitting it to you for your consideration. The reviewers’ comments and our corresponding responses are listed below.

Thank you very much for your kind attention on this matter.

Sincerely

Corresponding authors on behalf of all co-authors.

Luis M. Pérez-Belmonte, MD, PhD

María D. López-Carmona, MD, PhD.

luismiguelpb1984@gmail.com; mdlcorreo@gmail.com

Reviewer #2:

The authors describe a retrospective population-based study of all adult patients with diabetes who were hospitalized from 1997 to 2015 in Spain. Overall potentially preventable hospitalizations and hospitalizations by diabetes-related preventable conditions (short-term complications, long-term complications, uncontrolled diabetes and lower-extremity amputations) were investigated.

Data are well presented and thorough. One of the strengths of this study is the very long time considered in the analysis (19 years) as well as the wide sample of patients enrolled (whole Spanish country).

Among DM long-term complications the authors also cited eye complications. Have they also considered data from some diabetic eye screening programs?

Authors’ reply: At present, we have not considered including data from diabetic eye screening programs because we do not have full access to all national data. National health authorities are working on a full access program.